# Impact of Statin Use on Dementia Incidence in Elderly Men and Women with Ischemic Heart Disease

**DOI:** 10.3390/biomedicines8020030

**Published:** 2020-02-09

**Authors:** Mi-Young Kim, Minji Jung, Yoojin Noh, Sooyoung Shin, Chang Hyung Hong, Sukhyang Lee, Yi-Sook Jung

**Affiliations:** 1Department of Pathophysiology, College of Pharmacy, Ajou University, Suwon 16499, Korea; 2Division of Clinical Pharmacy, College of Pharmacy, Ajou University, Suwon 16499, Korea; abjeong@ajou.ac.kr (M.J.); eugenee81@gmail.com (Y.N.); syshin@ajou.ac.kr (S.S.); 3Department of Psychiatry, Ajou University School of Medicine, Suwon 16499, Korea; chhong2012@gmail.com

**Keywords:** statin, dementia, ischemic heart disease, sex difference

## Abstract

This study aimed to determine the association between statins and the prevention of dementia according to sex differences in elderly patients with ischemic heart disease (IHD). We performed a nationwide retrospective cohort study using the Korean Health Insurance Review and Assessment Service database (2007–2015). Among the 264,036 eligible patients aged ≥65 years with IHD, statin users were compared with non–users by propensity score matching at a 1:1 ratio (71,587 in each group). The primary outcome was dementia risk by estimating hazard ratios (HRs) and 95% confidence intervals (CIs). Differential risks of dementia were assessed by sex in the subgroups of statin types, exposure duration, and patient age, implying that sex is an influential factor for the link between statin use and dementia incidence. Among seven commonly prescribed statins, rosuvastatin was associated with the greatest preventive effect on dementia incidence, with an adjusted HR of 0.82 (95% CI = 0.78–0.87). In a subgroup analysis organized by sex, the differential risk of dementia incidence was assessed in each statin group, implying that sex is an influential factor for the link between statin and dementia. This study suggests that appropriate statin use considering sex differences may have beneficial effects on the development of dementia.

## 1. Introduction

Dementia, one of the major causes of cognitive decline among older people, is becoming a considerable burden on families and society due to the increasing life expectancy worldwide [1]. Patients with dementia numbered about 45 million worldwide in 2017 [2], and dementia is one of the leading causes of disability [3]. Therefore, dementia can increase healthcare burdens, while lowering the patients’ quality of life [4]. Dementia is a multifactorial neurodegenerative disorder with a number of risk factors, including age, dyslipidemia, stroke, and ischemic heart disease (IHD) [5], which provide potential targets for reducing the dementia risk among these patients and within the healthcare system [6].

Several studies have revealed important links between IHD and dementia [7,8,9,10]. The Bronx Aging Study showed that aged females with a history of IHD displayed a fivefold increase in the risk of dementia [9]. In addition, Ikram and colleagues demonstrated that IHD was associated with an increased risk of dementia [10]. Therefore, the development of an effective method for reducing the risk of dementia in the IHD population is clinically important.

The use of 3-hydroxyl-3-methylglutaryl coenzyme A reductase inhibitors, or statins, have long been recommended for people with cardiovascular disease, based on evidence from a number of clinical studies that demonstrate the lipid-lowering effects of statins in patients with IHD [11,12]. Recently, a number of reports have supported the notion that statins are associated with a reduced risk of dementia [13,14,15]. However, studies on the association between statin therapy and dementia risk are currently controversial. Randomized clinical trials have not drawn evident conclusions [16,17] and some studies have found no beneficial effects of statin therapy on dementia [11,18] due to the lack of a suitable sample size and insufficient follow-up durations. Furthermore, these studies have not focused on the population of patients with IHD. Therefore, whether statin use in these patients can produce beneficial effects on the risk of dementia remains unknown.

Although the importance of sex/gender differences has been demonstrated in human pathophysiology, most animal studies and clinical trials have been conducted on males [19,20], on the basis of the consideration that males are representative of human beings [21]. However, accumulated evidence shows sex-specific differences for many therapeutics in terms of efficacy and toxicity [22,23,24]. Moreover, a previous observational study demonstrated that the risk reduction of Alzheimer’s disease induced by statin use varied across sex and race/ethnicity [25]. A better understanding of sex-related differences in drug efficacy is essential for providing potential therapeutic strategies for men and women individually. Therefore, given the lack of research focusing on patients with IHD, the aim of this study was to investigate the association between statin use and dementia risk in elderly Korean patients with IHD across seven commonly prescribed statins in view of sex, age, and exposure duration differences.

## 2. Materials and Methods

### 2.1. Data Source

The data source was derived from the Korean Health Insurance Review and Assessment Service (HIRA) database for January 1, 2007 to December 31, 2015. The HIRA database contains medical information covering approximately 50 million Koreans. This information represents all patients’ health insurance claims, so we could obtain medical data such as demographic and socioeconomic information, disease diagnosis, prescribed medication information, and hospital and pharmacy visit information. The disease diagnosis was identified by using the International Classification of Diseases, tenth Revision (ICD–10) codes. To protect the privacy of patients, all individuals were de-identified, and patients in the database were given encrypted identification numbers. This study was approved by the Institutional Review Board of Ajou University (No. 201610–hb–EX–001).

### 2.2. Patient Population and Study Design

This study was a retrospective cohort study based on medical information gathered from the HIRA claim database. Elderly patients, aged 65 years and older, with a history of IHD, including Angina pectoris and acute/subsequent/chronic myocardial infarction (ICD–10; I20, I21, I22, I23, I24, and I25), from January 1 2007 to December 31 2015, were initially selected for the study cohort. Of these patients, those newly started on statin use in 2010 with a therapy duration of at least 6 consecutive months were included as statin users. We identified statins as seven commonly prescribed statins: atrorvastatin, simvastatin, rosuvastatin, pitavastatin, pravastatin, fluvastatin, and lovastatin. To identify new statin users, we only enrolled statin users with no history of statin treatment before 2010—the index year. A comparison cohort group of statin non-users was selected by random sampling, with adjustment for sex and age. Of the eligible populations selected, statin non-users were then matched at a 1:1 ratio with statin users using propensity score (PS). The index date was defined as the date of the first statin prescription record for statin users, and the first date of the medical record of the index year was defined as the index date for the comparison group—statin non-users. We defined new-onset dementia as a record of the diagnosis of dementia with the ICD–10 code of F00 (dementia in Alzheimer’s disease), F01 (vascular dementia), F02 (dementia in other diseases classified elsewhere), F03 (unspecified dementia), G30 (Alzheimer disease), and G31 (other degenerative diseases of the nervous system). To identify the incidence of new dementia, patients who had dementia diagnosis codes or were prescribed anti-dementia drugs prior to the cohort entry were excluded. Patients who started with two different types of statins at the same time were also excluded. The process of study population selection is shown in Figure 1.

### 2.3. Variables

We defined the age as the value at the year of the index date and classified it into three categories: 65–75, 76–85, and over 86 years. Comorbidities, including hypertension, diabetes, ischemic stroke, depression, Parkinson’s disease, and schizophrenia, were defined as those that had appeared more than two times between 1 year before the index date and 30 days after the index date. The Charlson Comorbidity Index (CCI) score was calculated to measure the comorbidity status [26] and classified into three categories: 0–1, 2, and over 3. Concurrent medications, including antiplatelet agents, angiotensin-converting enzyme inhibitors, angiotensin receptor blockers, beta-blockers, calcium channel blockers, proton pump inhibitors, and antidepressants, were considered during the study period. Concurrent medications were defined as those that been prescribed for at least 30 consecutive days.

### 2.4. Study Outcomes

The primary outcome was the incidence of new-onset dementia, depending on statin use and sex. Dementia incidence was defined as the earliest diagnosis record 6 months after the index date. The follow–up period was calculated from the index date to the date of dementia incidence, or the death date in the case of patients who died during the study period. If an outcome of dementia incidence did not occur, patients were censored at the end of the study, on December 31, 2015.

To assess the separate effects of short-term and long-term treatment for a subgroup analysis to minimize bias, patients in each statin group were divided into two subgroups, based on the duration of statin therapy (< 1 year and ≥ 1 year). To assess for age-differential outcomes linked to statin therapy for another subgroup analysis, statin group patients were categorized into three subgroups by patient age at the study entry (65–75 years, 75–85 years, and >85 years). The incidence and risk of new-onset dementia in accordance with statin therapy versus non-therapy were first assessed for all patients and then separately according to patient sex.

### 2.5. Statistical Analysis

Statin users and non-users were selected by random sampling with stratified sex and age in the index year, and then matched at a 1:1 ratio on the basis of PS matching, to reduce the imbalance of potential confounding factors. The PSs were estimated by fitting a multivariable logistic regression model, including baseline characteristics such as age, sex, and comorbidities. We compared all baseline characteristics between the PS-matched groups of statin users and statin non–users, using the chi-square test for categorical variables or t-test for continuous variables. The statistical significance of the *p*-value was designated as under 0.05. Moreover, using the Cox proportional regression model, we estimated hazard ratios (HRs) and 95% confidence intervals (95% CIs) between statin users and non-users, with non-users as the reference group. To reduce bias, we performed a multivariate analysis by adjusting the HRs for the following variables: a history of ischemic stroke, depression, and schizophrenia and all concurrent medications. Subgroup analyses stratified by sex and age were conducted to assess the risk of dementia. All data analyses were conducted using SAS Version 9.4 (SAS Institute, Cary, NC, USA).

## 3. Results

### 3.1. Baseline Characteristics

This cohort study included 2,408,821 elderly patients with IHD, who were aged 65 or older, from 2007 to 2015. Among those, 71,587 statin users and 71,587 matched non-users were gathered by applying the inclusion and exclusion criteria shown in Figure 1. Overall, there were 85,437 (60%) female patients and the total mean age was 72 years old. The median follow-up duration was 5.2 years in non-users and 5.0 years in statin users. The baseline characteristics for statin users and non-users are listed in Table 1.

### 3.2. The Impact of Statin Use on Dementia Risk in Men and Women with IHD

Table 2 shows the hazard ratios (HRs) for dementia incidence in statin users compared with non-users in men and women with IHD. In statin users, the most commonly prescribed statin type was atorvastatin (63.9%), followed by simvastatin (15.2%), rosuvastatin (11.5%), pitavastatin (3.7%), pravastatin (3.5%), fluvastatin (1.3%), and lovastatin (0.8%). Among these, rosuvastatin was associated with the greatest (18%) protective effect on dementia incidence, with an adjusted HR of 0.82 (95% CI, 0.78–0.87), whereas simvastatin and lovastatin showed no significant risk reduction for dementia. In the male group, total statin users had a significantly lower risk of dementia than statin non-users (adjusted HR, 0.92; 95% CI, 0.88–0.96). Among those, atorvastatin, rosuvastatin, and pravastatin users showed significantly decreased HRs for dementia (adjusted HRs, 0.92, 0.87, and 0.83, respectively). In the female group, total statin users also had a slightly lower risk of dementia compared with statin non-users (adjusted HR, 0.96; 95% CI, 0.93–0.99). In particular, rosuvastatin, pravastatin, and lovastatin users showed remarkable effects on decreasing the risk of dementia in the female group (adjusted HRs, 0.82, 0.89, and 0.74, respectively) (Table 2).

### 3.3. Association between Statin Use and Dementia Risk According to the Exposure Duration of Statin

As shown in Figure 2, for an assessment of the cumulative effects of long-term treatment by the duration of statin therapy, a tendency toward protective effects against new-onset dementia was observed with longer treatment (≥ 1 year) in five statin groups, when male and female patients were collectively assessed: the adjusted HRs (95% CI) associated with ≥ 1 year with therapy of atorvastatin, rosuvastatin, pitavastatin, pravastatin, and fluvastatin were 0.95 (0.92–0.98), 0.81 (0.76–0.86), 0.87 (0.78–0.96), 0.86 (0.78–0.95), and 0.83 (0.69–0.99), respectively. On the contrary, the risk for dementia incidence was found to be increased with short–term (< 1 year) use with atorvastatin and simvastatin: the adjusted HRs (95% CI) were 1.13 (1.05–1.22) and 1.16 (1.05–1.27), respectively. The adjusted HRs of the remaining five statins did not reach statistical significance.

In subsequent analyses, male and female patients categorized in each statin group were assessed separately. In male patients, those with long-term use for ≥ 1 year of atorvastatin, rosuvastatin, or pravastatin were less likely to develop dementia: the adjusted HRs (95% CI) were 0.91 (0.87–0.96), 0.85 (0.77–0.93), and 0.78 (0.66–0.93), respectively. None of the other subgroups in male patients showed a statistically significant difference in dementia risk compared with non-users. For female patients, the long-term use of atorvastatin, rosuvastatin, pitavastatin, or lovastatin appeared to reduce the risk of dementia incidence: the adjusted HRs (95% CI) were 0.96 (0.93–0.99), 0.81 (0.75–0.88), 0.86 (0.76–0.98), and 0.66 (0.49–0.89), respectively. On the other hand, short-term therapy with either atorvastatin or simvastatin was associated with an increased risk of dementia in female patients: the adjusted HRs (95% CI) were 1.18 (1.07–1.29) and 1.20 (1.07–1.35), respectively. Any other HRs in female subgroups did not show statistical significance.

### 3.4. Association between Statin Use and Dementia Risk According to Age

In general, statin effects tended to be either protective against dementia development or statistically insignificant relative to non-use. More specifically, for age, atorvastatin users in the subgroups of 75–85 years and > 85 years were less likely to develop dementia compared to non–users: the adjusted HRs (95% CI) were 0.89 (0.85–0.93) and 0.86 (0.74–0.99), respectively. When male and female patients were assessed separately, only the 75–85 years subgroup showed statistically significant HRs: 0.86 (0.79–0.94) and 0.89 (0.85–0.95) in male and female patients, respectively. Rosuvastatin users also showed a lower risk of dementia incidence in the subgroups of 65–75 years and 75–85 years: the adjusted HRs (95% CI) were 0.81 (0.75–0.87) and 0.81 (0.74–0.89), respectively. In female patients, dementia-protective effects were observed in all three subgroups: 0.82 (0.74–0.90), 0.81 (0.72–0.91), and 0.65 (0.43–0.97) in 65–75 years, 75–85 years, and >85 years, respectively, whereas in male patients, the dementia risk was lower in the two subgroups of 65–75 years and 75–85 years: 0.84 (0.75–0.94) and 0.85 (0.73–0.98), respectively. Pravastatin use appeared to decrease the overall risk of dementia in patients: the adjusted HRs (95% CI) were 0.86 (0.77–0.97) and 0.86 (0.74–0.99) in the subgroups of 65–75 years and 75–85 years, respectively; however, the statistical significance of the results disappeared when male and female patients were assessed separately. In most subgroups, the risk of dementia incidence was either lower or not significantly affected by the use of statins, except for the simvastatin group, where the adjusted HR (95% CI) associated with the 65–75 years subgroup was 1.08 (1.02–1.14) for all patients and 1.10 (1.03–1.18) in female patients (Figure 3).

## 4. Discussion

In this nationwide cohort study of Korea, we analyzed the association between statin use and dementia incidence in elderly patients with IHD, and have demonstrated that the association between statin use and dementia varies, depending on statin type, sex, exposure duration, and age. The main findings in elderly patients with IHD were as follows: (1) statin users exhibited a reduced risk of dementia incidence compared to non–users (adjusted HR 0.95, 95% CI 0.92–0.97); (2) among seven commonly prescribed statins, rosuvastatin was associated with the greatest (18%) protective effect on dementia incidence, with an adjusted HR of 0.82 (95% CI = 0.78–0.87); (3) in a subgroup analysis organized by sex, the differential risk of dementia incidence was assessed in each statin group, implying that sex is an influential factor for the link between statin and dementia; and (4) compared to no therapy, the long-term use of statins for ≥1 year generally tended to be more effective in preventing dementia than <1 year of therapy.

In this study, we examined the association between statin use and dementia incidence in elderly patients with IHD in view of sex, exposure duration, age, and types of statin. In the total group of patients, statin users exhibited significant protective effects against dementia incidence compared to non-users (adjusted HR 0.95, 95% CI 0.92–0.97), and atorvastatin, rosuvastatin, pitavastatin, pravastatin, and fluvastatin were significantly effective in reducing the dementia risk. In the separate analysis considering sex, the dementia-protective effects of statin were further confirmed with the adjusted HR (95% CI) of 0.92 (0.88–0.96) in male and 0.96 (0.93–0.99) in female patients. Rosuvastatin and pravastatin consistently showed decreased HRs in both males and females. Additionally, atorvastatin was associated with a reduced risk of dementia in males, but not in females, and lovastatin was associated with a reduced risk of dementia in females, but not in males.

Previous studies have reported that statins showed differential effects in reducing dementia risks according to ethnicity/race and sex; however, Asian patients were not sufficiently represented [25,27,28]. Consistent with our findings, the results from a Taiwan study have shown that among different types of statins, rosuvastatin was associated with a decreased risk of dementia in both sexes [28]. Interestingly, however, the results of atorvastatin and simvastatin in Taiwan patients were not consistent with our findings in Korean IHD patients, for which a significantly protective effect of atorvastatin was shown in both males and females, and that of simvastatin was shown in only females. In addition, several studies have further indicated a statin type-varying association between statin use and dementia incidence; however, they did not mainly focus on sex differences [11,18,28,29]. The inconsistency in the association between statin types and dementia incidence may be due to the lack of generalization of the study population. The diverse effects of statin on dementia incidence based on baseline characteristics according to sex can be found in the Appendix A. In males, the protective effects of dementia on statin therapy were confirmed, irrespective of patients’ history of hypertension or diabetes. However, statin therapy was associated with dementia risk reduction in patients without hypertension or diabetes in females. Additionally, in the subgroup analysis of statin exposure duration and sex, the long-term use of statins for ≥1 year (based on atorvastatin, rosuvastatin, pitavastatin, pravastatin, or fluvastatin) appeared to be more effective in preventing new-onset dementia than short-term use for < 1 year. Similarly, a previous meta-analysis that pooled data from eight studies with long-term (over 1 year) therapy demonstrated a 29% reduction in dementia incidence in statin-treated patients [30]. In the Taiwan study, statin therapy for more than 1 year was associated with a lower risk of dementia [31]. Atorvastatin and rosuvastatin in particular demonstrated efficacy in reducing the dementia risk in both male and female patients when therapy lasted for ≥ 1 year; however, caution is advised with atorvastatin and simvastatin as their use for < 1 year has been associated with a higher risk of dementia in female patients. An adverse event reactions database analysis of FDA supported these findings, showing that atorvastatin and simvastatin were associated with significantly more reports of cognitive impairments than others; although this may be temporary and not lead to dementia after the withdrawal of medication [32].

Furthermore, in our subgroup analysis stratified by age and sex, atorvastatin and rosuvastatin were again associated with the most protective effects in dementia; rosuvastatin was evaluated as effective in all age subgroups among female patients. Of note, however, was that patients treated with simvastatin appeared to be at a higher risk of new-onset dementia in some of our subgroup analyses, where patients were divided by treatment duration, age, and sex. In our study, the increased risk of dementia in simvastatin users was mainly driven by those in the 65–75 age group or short-term therapy group in females; such results appeared inconsistent with previous study findings [28]. However, several factors may have contributed to the above findings in the present study, which included only those high-risk patients aged ≥ 65 years with preexisting IHD, and simvastatin has been known to be associated with an increased risk of type 2 diabetes, typically known as a risk factor for dementia [5].

There are several possible explanations for our results. First, although no distinct mechanisms of statins exist to explain their effects on dementia protection, many studies have discussed their cholesterol-lowering effects and pleiotropic effects, including the beta-amyloid deposition, anti-thrombotic effects, or anti-inflammatory effects of statins related to dementia protection [16,33,34,35,36]. In addition, since IHD is a well-known risk factor for dementia, anti-hyperlipidemia, anti-platelet, and anti-hypertensive therapy have been suggested to reduce the risk of dementia [37,38]. These findings may help to clarify the beneficial effects of statins on dementia in statin users and non-users with a PS-matched baseline. Second, pathophysiological differences between females and males could be a possible reason, since some studies had remarked on the sex differences in statin effects [39,40,41]. Moreover, in some animal studies, statins might have a greater clinical efficacy among males due to differences in the rate of metabolism, depending on sex [42,43]. Several epidemiological studies have reported greater cholesterol reductions by statins among women than men [44,45]. Furthermore, cytochrome P450 system enzyme expression may vary by sex [46,47]. Therefore, between-sex differences in clearance, bioavailability, and the clinical effects may be possible. However, the causes of these sex-related differences are unclear. Future research on the different effects according to sex is needed. Third, we demonstrated that rosuvastatin and pravastatin had protective effects on the dementia risk in overall patients. A previous chemical research study demonstrated that rosuvastatin was likely to be associated with lowering the risk of cognitive impairment through anti-inflammatory responses by inhibiting the activated nuclear factor kappa B (NF–kB) signaling pathway [48]. In addition, pravastatin had a protective effect through decreasing cerebrospinal fluid apolipoprotein, which has modulating effects on brain neuronal function and structure [49].

Taking these findings together, statins can be considered as a potential treatment for patients with IHD aged ≥ 65 years to prevent dementia, in addition to lowering cardio-cerebrovascular risks. To the best of our knowledge, the present study is the first large-scale nationwide study of a Korean population to explore the impact of sex on the relationship between statin use and the risk of dementia in patients with IHD. Our study yielded several important findings. Statin use is associated with a reduced risk of dementia in patients with IHD. Sex was an effect-modifying factor in the relationship between statin types and the risk of dementia.

The strengths of our study include the use of a nationwide dataset, which contained data from a large sample of IHD subjects and enabled the long-term continuous tracing of dementia risk in statin users and non-users. Therefore, this study reflects the real-world health care situation in Korea. However, a careful interpretation of our findings is needed due to several limitations. First, residual confounders, including laboratory data, or data about the degree and severity of dementia in claims data, were not available, so cannot be exclusively ruled out. To minimize this inherent bias and increase the comparability between study groups in the present observational study, we matched statin users and non-users based on PSs and adjusted the dementia risk by applying multivariate analysis. Second, we did not differentiate between the various etiologies of dementia. However, our study might have some meaning because, worldwide, there is a much higher rate of diagnoses for unspecified or mixed types of dementia than pure dementia. Third, the character of lipophilicity of statins was not considered. Although investigators might favor the protective effects of statin with the characters of lipophilicity [32], both lipophilic statin (atorvastatin) and hydrophilic statins (rosuvastatin, pitavastatin, pravastatin, and fluvastatin) were associated with a reduced risk of dementia in patients with IHD in the present study. Our findings were in line with those of several previous studies indicating that the preventive effect of statins on dementia incidence was independent of the lipophilicity of statins [31,50]. Despite these limitations, the Korean large-scale administrative database presented a great chance for facilitating assessment of the dementia protection of medical interventions.

## 5. Conclusions

Our study demonstrated that statin therapy was associated with a reduction in dementia risk in Korean patients with IHD. The beneficial effects of statins were varied according to sex differences. This suggests that patients with IHD can reduce their risk of dementia using a particular statin. Clinician should consider the apparent differences in statin efficacy between sexes, as well as types of statins. The use of appropriate statin types with regard to sex differences may provide a relatively inexpensive way to reduce the burden of dementia.

## Figures and Tables

**Figure 1 biomedicines-08-00030-f001:**
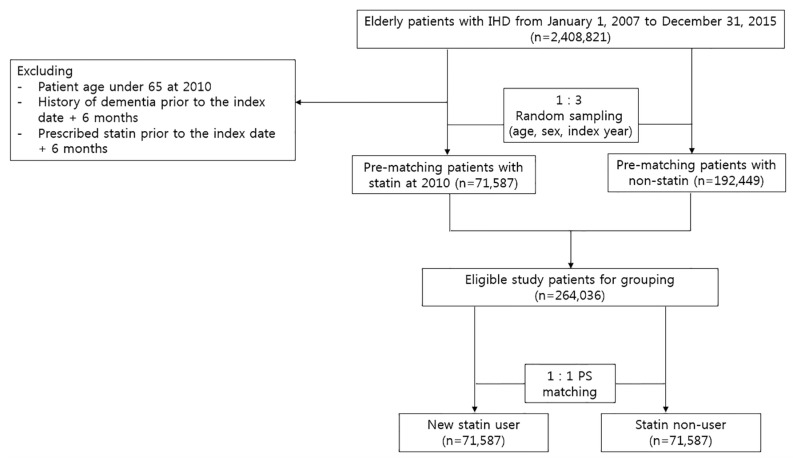
Flowchart of criteria used to select participants for the study.

**Figure 2 biomedicines-08-00030-f002:**
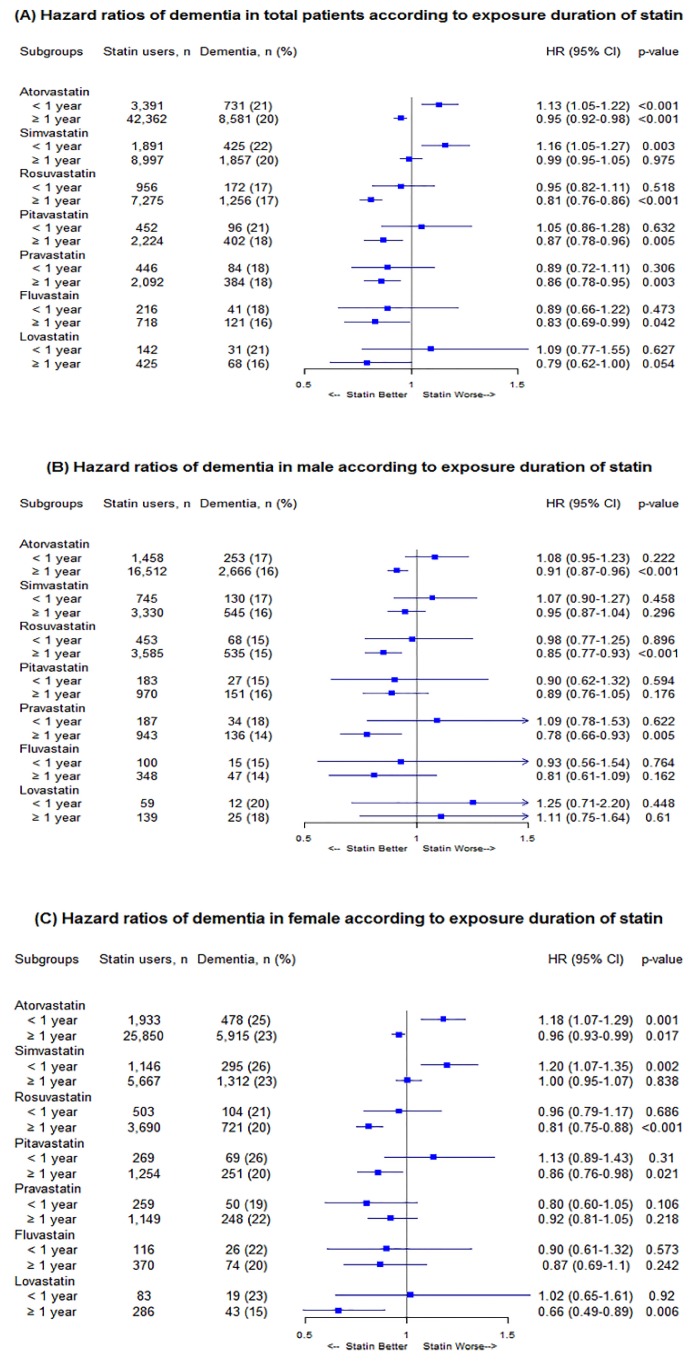
(**A**) Hazard ratios of dementia risk in total according to statin therapy duration, (**B**) hazard ratios of dementia risk in males according to statin therapy duration, and (**C**) hazard ratios of dementia risk in females according to statin therapy duration.

**Figure 3 biomedicines-08-00030-f003:**
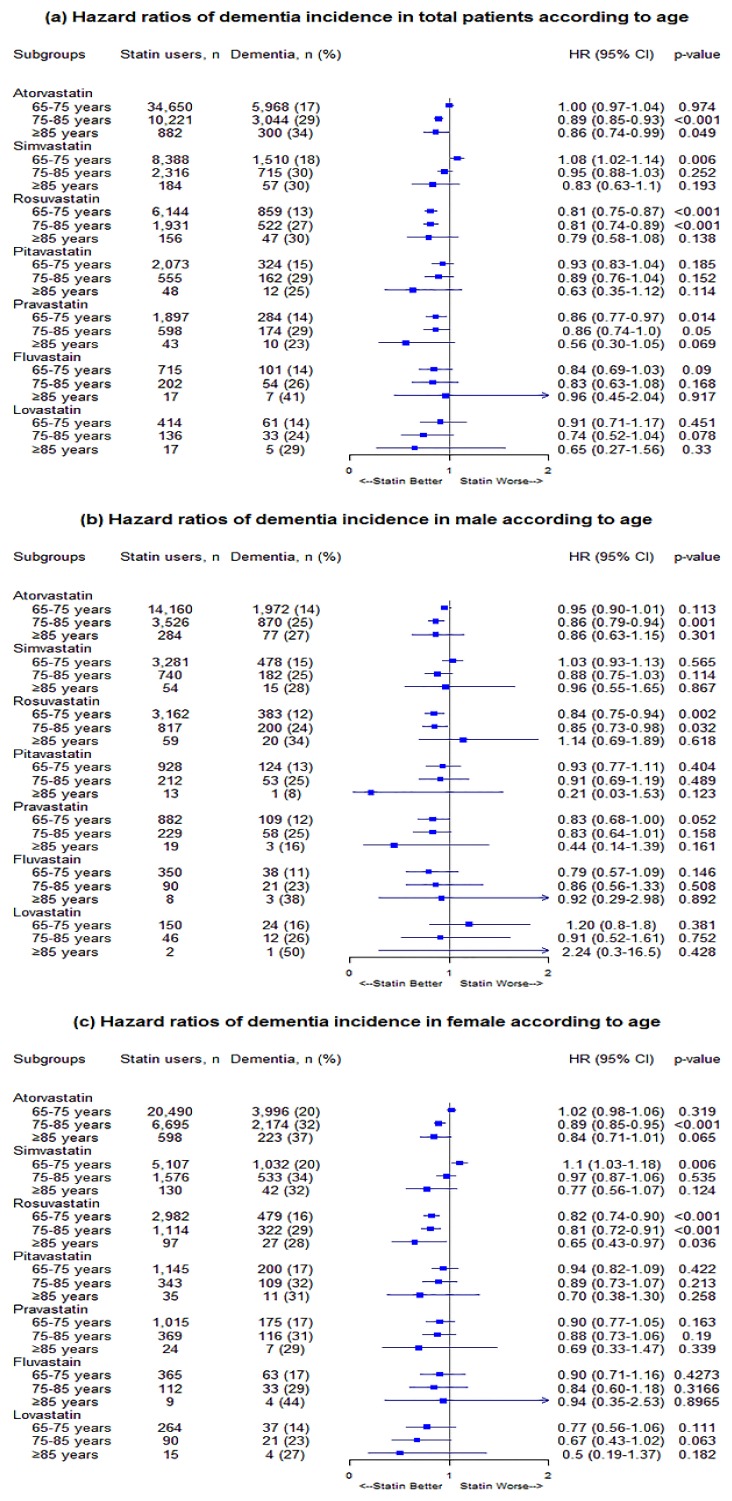
(**a**) Hazard ratios of dementia risk in total according to age, (**b**) hazard ratios of dementia risk in males according to age, and (**c**) hazard ratios of dementia risk in females according to age.

**Table 1 biomedicines-08-00030-t001:** Characteristics of study patients.

	Total (*n* = 143,174)		Male (*n* = 57,737)		Female (*n* = 85,437)	
	Non–Statin (*n* = 71,587)	Statin (*n* = 71,587)	Non–Statin (*n* = 28,725)	Statin (*n* = 29,012)	Non–Statin (*n* = 42,862)	Statin (*n* = 42,575)
	n (%)	n (%)	n (%)	n (%)	n (%)	n (%)
Age						
mean ± SD	72.3 ± 5.3	72.1 ± 5.4	71.7 ± 5.1	71.6 ± 5.2	72.6 ± 5.4	72.4 ± 5.5
65–75 years	54,292 (75.8)	54,281 (75.8)	22,685 (31.7)	22,913 (32.0)	31,607 (44.2)	31,368 (43.8)
75–85 years	16,014 (22.4)	15,959 (22.3)	5607 (7.8)	5660 (7.9)	10,407 (14.5)	10,299 (14.4)
>85 years	1281 (1.8)	1347 (1.9)	433 (0.6)	439 (0.6)	848 (1.2)	908 (1.3)
Comorbidities						
Hypertension	53,606 (74.9)	53,541 (74.8)	21,233 (29.7)	21,464 (30.0)	32,373 (45.2)	32,077 (44.8)
Diabetes	25,682 (35.9)	25,951 (36.3)	10,895 (15.2)	11,210 (15.7)	14,787 (20.7)	14,741 (20.6)
Ischemic stroke	8639 (12.1)	8935 (12.5)	4029 (5.6)	4334 (6.1)	4610 (6.4)	4601 (6.4)
Depression	6013 (8.4)	5720 (8.0)	1087 (1.5)	1783 (2.5)	4206 (5.9)	3937 (5.5)
Parkinson	767 (1.1)	712 (1.0)	287 (0.4)	263 (0.4)	480 (0.7)	449 (0.6)
Schizophrenia	262 (0.4)	137 (0.2)	130 (0.2)	62 (0.1)	132 (0.2)	75 (0.1)
CCI score						
≤1	30,328 (42.4)	30,319 (42.4)	11,600 (16.2)	11,607 (16.2)	18,728 (26.2)	18,712 (26.1)
2	15,887 (22.2)	15,816 (22.1)	6276 (8.8)	6296 (8.8)	9611 (13.4)	9520 (13.3)
≥3	25,372 (35.4)	25,452 (35.6)	10,849 (15.2)	11,109 (15.5)	14,523 (20.3)	14,343 (20.0)
Concurrent medication						
Antiplatelet agents	47,713 (66.7)	61,540 (86.0)	19,609 (27.4)	26,364 (36.8)	28,104 (39.3)	35,176 (49.1)
ACEIs or ARBs	43,101 (60.2)	52,675 (73.6)	17,200 (24.0)	21,614 (30.2)	25,901 (36.2)	31,061 (43.4)
Beta Blockers	25,479 (35.6)	36,279 (50.7)	9822 (13.7)	15,377 (21.5)	15,657 (21.9)	20.902 (29.2)
CCBs	42,245 (59.0)	43,666 (61.0)	16,367 (22.9)	16,673 (23.3)	25,878 (36.1)	26,993 (37.7)
Antidiabetic agents	21,770 (30.4)	26,268 (36.7)	9220 (12.9)	11,093 (15.5)	12,550 (17.5)	15,175 (21.2)
PPIs	30,867 (43.1)	37,237 (52.0)	12,043 (16.8)	14,423 (20.1)	18,824 (26.3)	22,814 (31.9)
Follow-up duration (year)						
mean ± SD	5.2 ± 1.7	5.0 ± 1.4	5.2 ± 1.8	5.0 ± 1.3	5.1 ± 1.7	4.9 ± 1.5

CCI, Charlson comorbidity index; ACEI, angiotensin-converting enzyme inhibitor; ARB, angiotensin II receptor blocker; CCB, calcium channel blockers; PPI, proton pump inhibitors.

**Table 2 biomedicines-08-00030-t002:** Impact of statin use on the dementia risk in patients with ischemic heart disease (IHD)**.**

	Dementia Incidence, n (%)				
	Total (*n* = 143,174)				
	*n*	case, *n*	(%)	HR (95% CI)	*p*-value
Non–user	71,587	14,122	19.7	reference	
Statin user	71,587	14,249	19.9	0.95 (0.92–0.97)	<0.001
Atorvastatin	45,753	9312	20.4	0.96 (0.93–0.99)	0.003
Simvastatin	10,888	2282	21.0	1.02 (0.98–1.07)	0.320
Rosuvastatin	8231	1428	17.3	0.82 (0.78–0.87)	<0.001
Pitavastatin	2676	498	18.6	0.89 (0.82–0.98)	0.013
Pravastatin	2538	468	18.4	0.86 (0.78–0.94)	0.001
Fluvastatin	934	162	17.3	0.84 (0.72–0.98)	0.032
Lovastatin	567	99	17.5	0.87 (0.71–1.06)	0.164
	Male (*n* = 57,737)				
	*n*	case, *n*	(%)	HR (95% CI)	*p*-value
Non–user	28,725	4619	16.1	reference	
Statin user	29,012	4644	16.0	0.92 (0.88–0.96)	< 0.001
Atorvastatin	17,970	2919	16.2	0.92 (0.88–0.97)	0.001
Simvastatin	4075	675	16.6	0.98 (0.90–1.06)	0.582
Rosuvastatin	4038	603	14.9	0.87 (0.80–0.95)	0.002
Pitavastatin	1153	178	15.4	0.90 (0.78–1.04)	0.157
Pravastatin	1130	170	15.0	0.83 (0.71–0.97)	0.016
Fluvastatin	448	62	13.8	0.85 (0.66–1.08)	0.188
Lovastatin	198	37	18.7	1.15 (0.84–1.60)	0.385
	Female (*n* = 85,437)				
	*n*	case, *n*	(%)	HR (95% CI)	*p*-value
Non–user	42,862	9503	22.2	reference	
Statin user	42,575	9605	22.6	0.96 (0.93–0.99)	0.003
Atorvastatin	27,783	6393	23.0	0.97 (0.94–1.00)	0.075
Simvastatin	6813	1607	23.6	1.03 (0.98–1.09)	0.275
Rosuvastatin	4193	825	19.7	0.82 (0.76–0.88)	<0.001
Pitavastatin	1523	320	21.0	0.90 (0.80–1.01)	0.063
Pravastatin	1408	298	21.2	0.89 (0.79–0.99)	0.048
Fluvastatin	486	100	20.6	0.87 (0.71–1.06)	0.164
Lovastatin	369	62	16.8	0.74 (0.58–0.95)	0.018

HR, hazard ratio; CI, confidence interval.

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
