# Peer review of "Impact of Statin Use on Dementia Incidence in Elderly Men and Women with Ischemic Heart Disease"

_biomedicines, 2020, doi:10.3390/biomedicines8020030_

Round 1
Reviewer 1 Report
The article is well-written, with only minor English mistakes. The manuscript contains all the necessary sections, that describe the methodology of the study, the statistics etc. However, the order of these sections seems inappropriate. The Material and methods should come immediate after the aim of the study, the statistical analysis after Materials and Methods. The References are new and appropriate.
Author Response
# Reviewer: 1
Comment: The article is well-written, with only minor English mistakes. The manuscript contains all the necessary sections, that describe the methodology of the study, the statistics etc. However, the order of these sections seems inappropriate. The Material and methods should come immediate after the aim of the study, the statistical analysis after Materials and Methods. The References are new and appropriate.
☞ Response: We have changed the order of section of Material and Methods and revised the number of references appropriately as you mentioned.
Reviewer 2 Report
First of all the authors should follow the recommended structure of an article: Material and Methods should go before Results, not after them.
Introduction and Discussion should show the existing controversies in the effect of statin use on dementia incidence: some studies showing positive, but some studies showing negative effect. Possible mechanisms of positive and negative effects could be mentioned.
Page 10 - several times incident dementia is mentioned. Probably it is dementia incidence?
Author Response
Point 1: Introduction and Discussion should show the existing controversies in the effect of statin use on dementia incidence: some studies showing positive, but some studies showing negative effect. Possible mechanisms of positive and negative effects could be mentioned. ☞ Response: Recently, a number of studies have shown the controversies in the effects of statin use on dementia incidence. There are previous studies showing positive effect with reduction of dementia incidence. There are some studies with non-positive or not significant effect on the incidence. But there are no study with negative effects.
We have mentioned those several studies and probable mechanisms and related reasons in sections of [INTRODUCTION] and [DISCUSSION] in the original manuscript: Introduction (line 68-74) and Discussion (line 260-261, line 268-270, line 283-286, line 297-317)
In addition, we have incorporated the reviewers’ suggestions in the revised manuscript for clarification. In section of [INTRODUCTION], we have added the statement about the possible reasons for no beneficial effects of statin use on dementia incidence as follows:
☞ Revision: Line 71-73. Randomized clinical trials have not drawn the evident conclusions [16–17] and some studies found no beneficial effects of statin therapy on dementia [11, 18] due to lack of sample size and insufficient follow-up durations.
Point 2: in Page 10, several times incident dementia is mentioned. Probably it is dementia incidence? ☞ Response: Yes, incident dementia means dementia incidence. We have changed the statement ‘incident dementia’ into ‘dementia incidence’ throughout manuscript.